# Exposure to Airborne Pesticides and Its Residue in Blood Serum of Paddy Farmers in Malaysia

**DOI:** 10.3390/ijerph19116806

**Published:** 2022-06-02

**Authors:** Siti Khairunnisaq Rudzi, Yu Bin Ho, Eugenie Sin Sing Tan, Juliana Jalaludin, Patimah Ismail

**Affiliations:** 1Department of Environmental and Occupational Health, Faculty of Medicine and Health Sciences, Universiti Putra Malaysia, Serdang 43400, Malaysia; nisaarudzi@gmail.com (S.K.R.); juliana@upm.edu.my (J.J.); 2School of Healthy Aging, Aesthetic and Regenerative Medicine, Faculty of Medicine and Health Sciences, UCSI University, Kuala Lumpur 56000, Malaysia; eugenietan@ucsiuniversity.edu.my; 3Department of Biomedical Science, Faculty of Medicine and Health Sciences, Universiti Putra Malaysia, Serdang 43400, Malaysia; patimah@upm.edu.my

**Keywords:** pesticides, inhalation exposure, personal air monitoring, blood serum, UHPLC-MS/MS

## Abstract

Background: Pesticides manage pests and diseases in agriculture, but they harm the health of agricultural workers. Concentrations of thirteen pesticides were determined in personal air and blood serum of 85 paddy farmers and 85 non-farmers, thereafter associated with health symptoms. Method: Samples were analyzed using ultra-high performance liquid chromatography-tandem mass spectrometry (UHPLC-MS/MS). Results: The median concentration of pesticides in personal air samples ranged from 10.69 to 188.49 ng/m^3^ for farmers and from 5.79 to 73.66 ng/m^3^ for non-farmers. The median concentration of pesticides in blood serum was from 58.27 to 210.12 ng/mL for farmers and 47.83 to 62.74 ng/mL for non-farmers. Concentration of eleven pesticides in personal air and twelve pesticides in blood serum were significantly higher in farmers than non-farmers (*p* < 0.05). All pesticides detected in personal air correlated significantly with concentration in the blood serum of farmers (*p* < 0.05). Health symptoms reported by farmers were dizziness (49.4%), nausea (47.1%), cough (35.3%), chest pain (30.6%), breathing difficulty (23.5%), sore throat (22.4%), vomiting (18.8%), phlegm (16.5%), and wheezing (15.3%). Concentration of pesticides in personal air, blood serum, and health symptoms were not significantly associated. Conclusion: Occupational exposure to pesticides significantly contaminates blood serum of farmers compared to non-farmers.

## 1. Introduction

Agriculture is a crucial economic sector in Malaysia. In 2019, the agriculture sector generated 7.1% of the national Gross Domestic Product (GDP) [1]. The paddy is a common agricultural crop in Malaysia for rice production, which is staple food for its citizens and neighboring countries. There are two seasons of paddy cultivation in Malaysia, with each season being four to five months. Each paddy season begins with preparation of agricultural fields and ends with crop harvest. 

The agricultural sector relies extensively on pest management to eliminate pests and diseases during crop production. Chemical pesticides are preferred, effective, and routinely used in agricultural fields [2]. While a major portion of sprayed chemical pesticides are dispersed into the environment, less than 0.1% play its intended role in pest management [3]. Pesticides remain in the environment and enter the food chain, causing deleterious health effects for human being. Commonly reported acute health problems associated with pesticides poisoning are wheezing, cough, irritation of respiratory tract, blood in sputum [4], burning eyes, blurred vision, skin irritation, excessive sweating, sore throat, shortness of breath, burning sensations in nose [5], headache, and dizziness [6].

Occupational exposure to pesticides occurs during the preparation of agricultural fields and the spraying of crops through inhalation or dermal contact [7,8,9]. Paddy farmers were reported to be continuously exposed to pesticides at levels significantly higher than the general population [10]. The Department of Standards Malaysia has developed a code of recommended practice incorporating guidelines for inhalation and dermal protection (MS 479:2012) [11]. However, only 8.4% of Malaysia’s paddy farmers in Tanjung Karang wore proper personal protective equipment (PPE). Hamsan et al. (2017) highlighted that most paddy farmers had adequate PPE for dermal protection but lacked PPE for inhalation [12]. 

There are limited studies on pesticide contamination in personal air samples, particularly the ‘currently used pesticides (CUPs)’. Most studies focused on older generation and persistent pesticides such as organochlorine (OC) and organophosphate (OP) groups [13,14,15,16]. CUPs have been preferred in recent years and are widely and rampantly used in Malaysia since they exhibit lower environmental persistence. Thus, this study aimed to investigate occupational exposure to the mixture of airborne pesticides, its contamination in farmers’ blood serum and associated health symptoms. Personal air samples and farmers’ blood serum were analyzed for 13 CUPs (azoxystrobin, buprofezin, chlorantraniliprole, difenoconazole, fipronil, imidacloprid, isoprothiolane, pretilachlor, propiconazole, pymetrozine, tebuconazole, trifloxystrobin, and tricyclazole). The target compounds were selected due to their popularity according to the interview conducted among paddy farmers in Tanjung Karang.

## 2. Materials and Methods

### 2.1. Sampling Methodology

The comparative study was conducted at Tanjung Karang, Selangor, Malaysia, an agricultural village with paddy cultivations. It has 24 paddy blocks consisting of paddy fields and farmers’ residential houses. A total of 85 paddy farmers and 85 non-farmers were monitored and assessed in this study as exposed and comparative groups respectively. Inclusion criteria for the exposed group were (i) farmers involved in the preparation and application of pesticides; (ii) aged between 18 and 59 years old; and (iii) male. Meanwhile, inclusion criteria for the comparative group were (i) healthy male adults residing within the same area as the exposed group; (ii) aged between 18 and 59 years old; and (iii) had no prior history of occupational exposure to pesticides. Only male farmers were included in this study since all of the farmers working in the paddy fields in Kampung Sawah Sempadan were male. 

### 2.2. Questionnaire Data

This research was conducted following the Declaration of Helsinki. Ethical approval for this study was granted by the Ethics Committee for Research Involving Human Subjects Universiti Putra Malaysia (JKEUPM (FPSK-P161) 2017. Farmers and non-farmers were administered with questionnaires via convenient sampling to obtain demographics, exposure to pesticides, occupational exposure, and health information. Questionnaires were developed based on previously published studies such as (i) Agriculture Health Study [17,18,19] and (ii) Vietnam: Pesticide Use Survey [20].

### 2.3. Personal Air Samples

Personal air sampling for farmers coincided with mixing, loading, and spraying of pesticides. Meanwhile, non-farmers were sampled during their working hours. Duration for personal air sampling was the same for both groups. Exposure to pesticides was monitored using a personal air sampler coupled with an air pump (Escort^®^ Elf Air Sampling Pump, Zefon International Inc, Ocala, FL, USA) and a solid sorbent tube (SKC Sorbent Tube, XAD-2, 8 × 110mm size, 200/400 mg sorbent, SKC, Eighty Four, PA, USA). An air sampler was attached within the respondent’s breathing zone. The airflow rate was set at 2 L/min. Methods for sampling and extraction of personal air samples were in accordance with a study by Choi, Moon, and Kim (2013) [21]. Chemicals used for extraction are listed in online Appendix A. After sampling, sorbent tubes were capped and sealed using parafilm to prevent cross-contamination. After that, they were wrapped in aluminum foil, packed in a zip-lock bag, and stored at −20 °C until further extraction. The sorbents were extracted with 10 mL of acetone and spiked with 50 ppb of imidacloprid-d_4_ as internal standard (IS). They were vortexed for 1 min and centrifuged at 40 × 100 rpm for 5 min. Extracts were concentrated under a stream of nitrogen gas and reconstituted with 1 mL of injection solution (3:1, Ultrapure water: HPLC-grade methanol). Then, they were analyzed using ultra-high performance liquid chromatography-tandem mass spectrometry (UHPLC-MS/MS). 

### 2.4. Blood Serum Samples

Seven milliliters of blood samples were collected in a serum tube (BD Vacutainer Plus Plastic Serum Tube, Benton Dickson, Franklin Lakes, NJ, USA) using venipuncture by the medical doctor in a clinical setting. Blood samples sat at room temperature for 30 to 60 min for clotting. Thereafter, samples were centrifuged at 40 × 100 rpm for 10 min to obtain blood serum. Serums were kept in vials, sealed with parafilm, and stored at −20 °C until further analysis. Blood serums were extracted using the QuEChERS method by Shin et al. (2018) [22]. Chemicals used for extraction are listed in online Appendix A. Four hundred microliters of acetonitrile and 50 ppb of IS were added to 100 µL of serum samples. Then, they were shaken using a thermos-shaker at 1200 rpm for 1 min. Forty milligrams of magnesium sulphate and 10 mg of sodium chloride were added under ice bath. Tubes were centrifuged at 13,000 rpm for 5 min and supernatants were dried under a stream of nitrogen gas. Four hundred microliters of reconstitution solution (3:1, Ultrapure water: HPLC-grade methanol) were added to samples and analyzed using UHPLC-MS/MS. 

### 2.5. UHPLC-MS/MS Analysis

All samples were analyzed using UHPLC-MS/MS (Agilent, Santa Clara, CA, USA). Detailed methodology for UHPLC-MS/MS and its mobile phase gradients were explained in Appendix A, respectively.

### 2.6. Quality Assurance (QA) and Quality Control (QC)

UHPLC-MS/MS performance for all target compounds is shown in Table 1. Method detection limit (MDL) and method quantification limit (MQL) were determined by the signal-to-noise ratio of >3 and >10 respectively [23,24]. A five-point calibration curve was used to determine the instrument’s linearity with points that range from instrumental quantification limit (IQL) to 500 ng/mL. Extraction recoveries were calculated using Equation (1): (1)$$\mathrm{Recovery}\;\left(\%\right)=\frac{C_{\mathrm{spiked}}-{\;C}_{\mathrm{unspiked}}}{C_{\mathrm{added}}}\times 100$$

C_spiked_ is the concentration of pesticide quantified in the sample spiked with a mixture of native standards and IS, C_unspiked_ is the concentration of pesticide quantified in blank sample, and C_added_ is the known pesticide concentration that was added to the sample.

Pesticides in air samples were calculated using Equation (2) [25]:(2)$$C_{\mathrm{air}}=\frac{C\;\times V}{V_{\mathrm{air}}}$$

C_air_ is the concentration of pesticide quantified in an air sample (ng m^−3^), C is the concentration of pesticide quantified in extract (ng mL^−1^), V is the final volume of extract (1 mL), and V_air_ is the volume of air sampled (m^3^).

Derivations of air volume sampled (V) were calculated Equation (3) [25]:(3)$$V_{\mathrm{air}}=F\;\times T\;\times \mathrm{CF}$$

V_air_ is the volume of air sampled (m^3^), F is the flow rate (L min^−1^), T is the sampling duration (min), and CF is the conversion factor from liter to a cubic meter (0.001).

Concentration of pesticides in blood serum were calculated using Equation (4): (4)$$C_{\mathrm{blood}}=\frac{C\;\times V}{V_{\mathrm{blood}}}$$

C_blood_ is the concentration of pesticide quantified in blood serum (ng/mL), C is the concentration of pesticide quantified in extract (ng/mL), V is the final volume of extract (100 µL), and V_blood_ is the volume of blood serum (7 mL).

### 2.7. Statistical Analysis

Descriptive analysis was performed to describe respondents’ socio-demographic, pesticide exposure (exposure time, exposure frequency, exposure duration), concentration of pesticides in personal air samples, concentration of pesticides in blood serum, climatological conditions (wind speed, temperature, relative humidity), use of PPE, and personal hygiene among farmers, as well as their self-reported health symptoms. The significant differences in concentration of pesticides within personal air samples and blood serum samples were determined using the Mann-U Whitney test. Correlations between concentration of pesticides in air samples, concentration of pesticides in blood serum and climatological conditions, and concentration of pesticides in blood serum and Body Mass Index (BMI) were determined using the Spearman correlation coefficient test. Multiple logistic regressions were used to determine relationships between concentration of pesticides in personal air samples, blood serum, and self-reported health symptoms. Statistical data were performed using Statistical Package for the Social Sciences (SPSS) version 25 (IBM Corp, New York, USA).

## 3. Results

Table 2 shows respondents’ socio-demographics. All respondents recruited were male, a representative gender for most farmers involved in paddy cultivation. Respondents’ age ranged from 18 to 59 years old. Body mass index (BMI) for farmers were 2.4% underweight, 47.0% normal, 25.9% overweight, and 24.7% obese. Meanwhile, 31.8% of non-farmer respondents had normal BMI, 38.8% were overweight, and 29.4% were obese. Most farmers (84.7%) had secondary education, while only 2.4% had tertiary education. Meanwhile, 42.4% of non-farmers had secondary education, and 54.1% had tertiary education. Half of the respondents in each group were smokers. 

Table 3 describes farmers’ exposure to pesticides obtained via questionnaire. Exposure time was the average exposure hours daily, exposure frequency was the average exposure days yearly, and exposure duration was the average exposure years to pesticides. The average exposure time for paddy farmers engaged in pesticides spraying was 4 h. Meanwhile, the average exposure frequency was 169 days per year and the average exposure duration was 17 years. 

Table 4 summarizes the concentration of pesticides quantified in personal air samples for farmers and non-farmers. At least one pesticide was quantified in the personal air samples among farmers. The highest median concentration detected in personal air samples for farmers was isoprothiolane at 188.49 ng/m^3^, while tricyclazole was detected to have the lowest median concentration at 10.69 ng/m^3^. Meanwhile, imidacloprid was the highest median concentration in air samples at 73.66 ng/m^3^, and azoxystrobin was the lowest median concentration at 5.79 ng/m^3^ among non-farmers. Chlorantraniliprole, fipronil, isoprothiolane, pretilachlor, propiconazole, and trifloxystrobin were not detected among non-farmers. The concentration of buprofezin, chlorantraniliprole, difenoconazole, fipronil, isoprothiolane, pretilachlor, propiconazole, pymetrozine, tebuconazole, tricyclazole, and trifloxystrobin were significantly higher among farmers than non-farmers (*p*-value < 0.05). 

Table 5 shows farmers’ time-weighted average (TWA) of pesticides exposure and permissible exposure limit (PEL). This study’s time-weighted average of pesticides exposure ranged from 5.75 to 15.96 ng/m^3^, lower than the recommended 8-h TWA PEL (3.50 × 10^4^–8.00 × 10^6^ ng/m^3^).

Table 6 reports the concentration of pesticides detected in the blood serum of farmers and non-farmers. Among farmers, the highest median concentration was azoxystrobin at 210.12 ng/mL, while the lowest median concentration was tricyclazole at 58.27 ng/mL. Meanwhile for non-farmers, the highest median concentration was difenoconazole at 62.74 ng/mL and the lowest median concentration was fipronil at 47.83 ng/mL. Azoxystrobin, chlorantraniliprole, isoprothiolane, pretilachlor, propiconazole, pymetrozine, and trifloxystrobin were not detected in the blood serum of non-farmers. All pesticides except tricyclazole were significantly higher in the blood serum of farmers with a *p*-value < 0.05. 

Concentration of pesticides in personal air samples correlated significantly with pesticides in blood serum of farmers (*p* < 0.001) (Table 7). Imidacloprid had the strongest association (r = 0.937) followed by isoprothiolane (r = 0.917), chlorantraniliprole (r = 0.889), pretilachlor (r = 0.868), fipronil (r = 0.748), propiconazole (r = 0.746), difenoconazole (r = 0.745), azoxystrobin (r = 0.727). Buprofezin, pymetrozine, tebuconazole and trifloxystrobin showed moderate correlations (r = 0.556–0.656), while tricyclazole showed weak correlation (r = 0.367). In the non-farmers group, significant correlations were observed for buprofezin, difenoconazole, imidacloprid, and tebuconazole with moderate associations (r = 0.494–0.584) (*p* < 0.001).

The median wind speed recorded for farmers and non-farmers was 0.30 m/s and 0.20 m/s respectively. The median ambient temperature recorded for farmers and non-farmers was 28.9 °C and 29.1 °C, respectively. Median relative humidity for farmers was 90% and 66% for non-farmers (Appendix A). The results demonstrated significant correlations between chlorantraniliprole (r = −0.342), propiconazole (r = 0.252), and tricyclazole (r = 0.219) in personal air samples and wind speed (*p* < 0.001). Additionally, chlorantraniliprole in personal air samples had significant correlations with temperature (r = −0.463) and relative humidity (r = −0.435) with *p*-values <0.001 (Table 7). On the other hand, no significant correlation was found between pesticides concentration in blood serum and BMI (Table 7).

During observation, most farmers were reported not wearing appropriate PPE when spraying pesticides onto crops; 92.9% used old cloth as a face mask, 5.9% used a respirator, and 1.2% did not use any form of protection. All farmers covered their bodies with long-sleeved shirts and long pants. Additionally, 2.4% had waterproof coveralls, 4.7% had waterproof pants and 4.7% had a waterproof apron. Besides, 78.8% of farmers had boots or footwear at work. Most farmers (82.4%) also had caps or headcovers. Only 27.1% of farmers used gloves, and 9.4% used goggles during the preparation and spraying of pesticides. Notwithstanding the lack of compliance to proper PPE, all farmers reported good personal hygiene practices such as immediate cleaning and changing of clothes after spraying. Information on PPE use, and personal hygiene practices are elaborated in Appendix A. Farmers’ reasoning for neglecting proper use of PPE was recorded in Appendix A.

Respondents’ self-reported health symptoms included breathing difficulty, chest pain, cough, phlegm, wheezing, sore throat, nausea, vomiting, and dizziness (Appendix A). Among farmers, the most common reported symptom was dizziness (49.4%), followed by nausea (47.1%), cough (35.3%), chest pain (30.6%), breathing difficulty (23.5%), sore throat (22.4%), vomiting (18.8%), phlegm (16.5%), and wheezing (15.3%). Meanwhile, non-farmers were reported to have cough (32.9%), sore throat (16.5%), nausea (5.9%), dizziness (4.7%), phlegm (4.7%), vomiting (3.5%), breathing difficulty (2.4%), and chest pain (1.2%).

Multiple logistic regression analysis showed no significant association between concentration of pesticides in personal air samples, concentration of pesticides in blood serum and farmers’ self-reported health symptoms (*p* > 0.05). The association was adjusted for the use of PPE among farmers to determine the effect of confounder. Likewise, no significant association was found between concentration of pesticides in personal air samples, concentration of pesticides in blood serum, and farmers’ self-reported health symptoms (Appendix A). 

## 4. Discussion

In Malaysia, pesticides are subsidized by the government to boost agriculture output and income. However, extensive and uncontrolled use of pesticides is a health concern as more than 95% of applied pesticides disperse into the environment affecting non-target organisms [36]. Farmers are exposed to pesticide residues during handling and spraying onto crops. Spray drift refers to the diffusion of pesticides resulting in off-target contamination on farmers’ bodies and the surrounding environment [37].

All farmers’ personal air samples were detected to have at least one pesticide with median concentration which ranged from 10.69 ng/m^3^ to 188.49 ng/m^3^. Non-farmers were reported to have median concentration of pesticides from 5.79 ng/m^3^ to 73.66 ng/m^3^ in their personal air samples. Fifty-three out of eighty-five (62.4%) personal air samples were detected to have more than one pesticide. Results reflected the practice and preference for cocktail pesticides as they save time, energy and cost. A similar study by Hamsan et al. (2017) reported pesticide concentration up to 462.50 ng/m^3^ in farmers’ personal air samples from the same study area [12]. Baharuddin et al. (2011) detected 38.0 ng/m^3^ of paraquat and 56.0 ng/m^3^ of 2,4-D in personal air samples of pesticide sprayers in Kerian, Malaysia [3]. All TWA concentration of pesticides in personal air samples of farmers did not exceed the recommended PEL except for pretilachor and tebuconazole, of which comparison was not possible due to the absence of their PEL. Hence, pesticide exposure for farmers in this study was considered acceptable. Several studies reported comparable results of pesticides exposure below PEL [3,38]. 

Pesticides residue in the blood reflects the body’s exposure and burden for measured compounds [39]. The median concentration of pesticides quantified in farmers’ blood serums was between 58.27 ng/mL and 210.12 ng/mL. These concentrations were comparable to a study by Moshou et al. (2020), which found five pesticide residues (trifluralin, chlorpyriphos methyl, metolachlor, fenthion, and dimethoate) and three metabolites (fenthion sulfone, fenthion sulfoxide, and 4,4′ DDE) in blood serum of Greece farmers; of which their concentration ranged from 0.40 ng/mL to 48.00 ng/mL) [40]. Leili et al. (2022) reported mean concentration of dichlorvos, diazinon, and chlorpyrifos to be 350 ng/mL, 110 ng/mL, and 140 ng/mL, respectively, in the blood serum of greenhouse workers [41]. Several studies reported concentration of pesticides in blood to be lower than our findings. Herin et al. (2011) detected fipronil in blood serums of drug factory workers with a mean concentration of 0.47 ng/mL [42]. Meanwhile, Hayat, Ashfaq, Ashfaq, and Saleem (2010) detected pyributicarb and chlorpyrifos in the blood of Pakistanis pesticide applicators at 1.00 ng/mL and 9.00 ng/mL respectively [43].

All pesticides quantified in personal air samples were significantly higher in farmers than non-farmers except for azoxystrobin and imidacloprid. Likewise, 12 out of 13 pesticides (except tricyclazole) were significantly higher in the blood serum of farmers. Occupational exposure to pesticides is typically higher than environmental exposure. Thus, the risk of occupationally exposed groups is higher than normal population [7]. Likely, agricultural workers were exposed to pesticides during handling and spraying due to their close contact. However, Doğanlar et al. (2018) reported a higher concentration for those residing in Turkey’s agricultural area despite not being occupationally exposed compared to residents in the control area [44]. 

All pesticides quantified in farmers’ personal air samples correlated significantly with those in blood serum. Similarly, four out of five detected compounds (buprofezin, difenoconazole, imidacloprid, tricyclazole) correlated significantly in non-farmers personal air and blood serum. Previous studies by [45] reported similar findings where significant correlations were found between chlorpyrifos, diazinon, and propoxur in personal air and their corresponding concentrations in blood. These associations suggest absorption of externally exposed pesticides into the bloodstream and circulating them throughout the body. Inhaled pesticides are absorbed by the throat, nasal passages, lungs, and thereafter enter bloodstream. Respiratory exposure to pesticides is a significant concern due to the volatile characteristics of pesticides and their rapid absorption into the respiratory tract [8,46].

Climatological condition is an important factor affecting the presence and distribution of pesticides in the air. Wind speed measured at paddy field was lowest in the early morning and highest in the evening. Temperature also showed a similar pattern which nadirs during early morning and peaks during the afternoon. Malaysia’s sunny weather with dry season recorded high temperatures during our monitoring. On the contrary, relative humidity was higher during the early morning and decreased with increasing temperature. Chlorantraniliprole, propiconazole, and tricyclazole in personal air samples correlated significantly with wind speed, while chlorantraniliprole correlated significantly with temperature and relative humidity. Our findings is comparable with results by Hamsan et al. (2017) that found significant correlations between pesticides in air samples with wind speed (pymetrozine: r = −0.217, *p* = 0.049; fipronil: r = −0.231, *p* = 0.036) and temperature (chlorantraniliprole: r = −0.224, *p* = 0.041) [12]. Additionally, Baharuddin et al. (2011) reported that wind speed correlated significantly with the inhalation of pesticides by respondents with improper use of PPE (r = 0.33, *p* = 0.01) [3]. Given significant correlations between pesticides in air and climatological conditions, it is crucial to consider these factors when deciding the timing to spray crops. Additionally, pesticide sprayers should also avoid spraying when wind speeds are less than three mph (1.34 m/s) or more than ten mph (4.47 m/s) as temperature inversion can occur [37]. 

During sampling, high ignorance towards PPE among farmers was observed. Most farmers did not use appropriate PPE. Instead, they preferred making their face masks from old cloth as protective barrier (Appendix A). Most farmers mentioned it is uncomfortable to use an appropriate respirator (83.5%) due to their high energy-demanding activities and the need to carry heavy sprayers. Farmers added that using a piece of cloth is comfortable and economical as they reuse their worn-out clothing. Other reasons for this were lack of knowledge on the importance of using appropriate PPE (7.1%), the expensive cost of respirators (5.9%), and a lack of guidance on the correct use of a respirator (3.5%). A study by Choudhary et al. (2014) on the effects of pesticide use among sprayers in Bhopal, India, reported that none of them use PPE when handling the pesticides [5]. Similarly, alarming incidences of agriculture workers failing to use proper PPE had been reported in Ethiopia (93.2%) [47], Malaysia (91.6%) [12], and Brazil (41.7%) [48]. Although PPE is a required safety precaution against pesticide-related health hazards, it is regularly omitted and overlooked by farmers. 

Exposure to pesticides had been linked to various health effects, ranging from temporary acute effects such as eye and skin irritations [5] to severe chronic health concerns such as cancers [49,50] and reproductive disorders [51]. Some pesticides can bio-accumulate, causing prolonged chronic effects. However, disease-exposure relationships are challenging to establish due to the complexity of diseases and myriad contributing factors [52]. Alongside, Fuad et al. (2012) reported farmers’ complaints of breathing difficulties during and after spraying of pesticides (51.5%), itchiness and soreness (26%), as well as rashes and peeling of skin on hand (13.7%) [53]. To a more severe extent, there were incidences of farmers collapsing, experiencing stomach aches, vomiting, and admitting to the hospital [53]. Respiratory symptoms such as coughing, wheezing, and airway inflammations were regularly reported by those exposed to pesticides [9]. Prevalence of health symptoms in this study was higher among farmers than non-farmers postulating that exposure to pesticides could be an underlying environmental factor for these health concerns. 

An increased risk of chronic cough and shortness of breath (OR = 3.15, 95% CI = 1.56–6.36 and OR = 6.67, 95% CI = 2.60–17.58) was reported among farmers compared to the non-exposed group in Ethiopia [54]. In another study, it was found that wheezing among commercial pesticide applicators were significantly associated with exposure to chlorpyrifos (OR = 1.27, 95% CI = 0.92–1.74), dichlorvos (OR = 2.48, 95% CI = 1.08–5.66) and phorate (OR = 2.35, 95% CI = 1.36–4.06) [55]. However, this study did not find any significant association between concentration of pesticides in personal air samples, pesticides in blood serum and self-reported health symptoms. These results are subjected to respondents’ recall bias during the interview. In addition, health symptoms are self-reported by respondents with no verification of medical records. Additionally, pesticides studied were currently used pesticides (CUPs) that are less persistent than previous generation pesticides such as organochlorine and organophosphate. CUPs can be mobilized and eliminated from the body before they can cause health effects to farmers. Likewise, Hamsan et al. (2017) did not report any significant association between pesticide exposure and paddy farmers’ health symptoms [12].

## 5. Conclusions

Most pesticides in personal air samples and blood serums were significantly higher in farmers than non-farmers. The concentration of pesticides in personal air samples correlated significantly with pesticides in blood serums for both farmers and non-farmers (except tricyclazole), thereby suggesting relationships between external exposure and internalization into the human body. High incidences of health symptoms were reported among farmers but no significant association between concentration of pesticides in personal air samples, blood serums and self-reported health symptoms were found. This study calls for future research probing into the combination of pesticides exposure routes via inhalation, dermal pathways, and ingestion to emulate comprehensive real-life exposure. We also suggest that future researchers incorporate biomarkers of exposure as in-depth assessments of the effect of pesticides on farmers’ health. 

## Figures and Tables

**Table 1 ijerph-19-06806-t001:** Information on UHPLC-MS/MS and method performance.

Target Compounds	Linear Range (ng/mL)	R^2^	Recovery % (RSD %), *n* = 3		MDL		MQL	
			Personal Air (250 ng/Sample)	Blood Serum (250 ng/mL)	Personal Air (ng/sample)	Blood Serum (ng/mL)	Personal Air (ng/sample)	Blood Serum (ng/mL)
Azoxystrobin	0.1–500	0.9998	96.5 (4.0)	95.4 (6.6)	0.1	1.0	1.0	10.0
Buprofezin	0.1–500	0.9995	93.7 (4.4)	87.2 (3.2)	0.3	3.0	0.5	10.0
Chlorantraniliprole	1.0–500	0.9993	102.5 (9.2)	82.9 (13.8)	0.3	3.0	2.0	12.0
Difenoconazole	1.0–500	0.9997	97.2 (13.3)	92.1 (10.9)	0.5	3.0	2.5	10.0
Fipronil	0.5–500	0.9997	91.8 (6.1)	88.1 (4.3)	0.3	1.0	0.7	9.0
Imidacloprid	0.1–500	0.9996	98.9 (8.8)	86.3 (6.3)	0.2	1.0	0.7	6.0
Isoprothiolane	0.1–500	0.9997	78.9 (8.50	84.3 (7.2)	0.1	1.0	0.5	8.0
Pretilachlor	1.0–500	0.9999	90.4 (1.8)	83.0 (10.3)	0.5	3.0	2.0	9.0
Propiconazole	0.5–500	0.9995	88.4 (5.7)	82.2 (2.8)	1.0	3.0	3.0	11.0
Pymetrozine	0.1–500	0.9999	100.6 (3.5)	90.0 (11.9)	0.3	5.0	0.6	12.0
Tebuconazole	0.1–500	0.9999	91.0 (12.3)	78.9 (7.3)	0.3	1.0	0.5	8.0
Tricyclazole	0.1–500	0.9991	104.7 (6.2)	103.4 (3.6)	0.1	3.0	0.5	12.0
Trifloxystrobin	0.1–500	0.9999	99.3 (6.1)	91.8 (3.6)	0.5	1.0	0.7	9.0

**Table 2 ijerph-19-06806-t002:** Socio-demographic information of the respondents.

Variables Groups	Mean ± SD		
	Famers (*n* = 85)	Non-Farmers (*n* = 85)	
Age (years)	42.54 ± 11.06	40.92 ± 11.18	
Variables	Categories	Frequency, *n* (%)	
Groups		Farmers (*n* = 85)	Non-farmers (*n* = 85)
Gender	Male	85 (100.0)	85 (100.0)
	Female	0 (0)	0 (0)
Race	Malay	85 (100.0)	85 (100.0)
	Chinese	0 (0)	0 (0)
	Indian	0 (0)	0 (0)
BMI	Underweight	2 (2.4)	0 (0)
	Normal	40 (47.0)	27 (31.8)
	Overweight	22 (25.9)	33 (38.8)
	Obese	21 (24.7)	25 (29.4)
Educational background	No formal education	0 (0)	0 (0)
	Primary	11 (12.9)	3 (3.5)
	Secondary	72 (84.7)	36 (42.4)
	Tertiary	2 (2.4)	46 (54.1)
Smoking Status	Yes	43 (50.6)	43 (50.6)
	No	42 (49.4)	42 (49.4)

**Table 3 ijerph-19-06806-t003:** Pesticides exposure information of paddy farmers (*n* = 85).

Variables	Average	Minimum	Maximum
Exposure time (hour/day)	4	1	12
Exposure frequency (day/year)	169	32	224
Exposure duration (years)	17	1	40

**Table 4 ijerph-19-06806-t004:** Concentration of target compounds detected in personal air samples of farmers and non-farmers.

Target Compounds	Median ^a^ (IQR), (ng/m^3^)		Frequency of Detection, *n* (%)		*p*-Value	*Z*-Score
	Farmers	Non-Farmers	Farmers	Non-Farmers		
Azoxystrobin	15.53 (12.17–36.97)	5.79 (4.20–10.04)	12 (14.1)	6 (7.1)	0.150	−1.439
Buprofezin	45.15 (16.12–74.56)	19.05 (10.82–28.57)	28 (32.9)	8 (9.4)	<0.001	−3.957
Chlorantraniliprole	78.94 (50.99–165.29)	ND	23 (27.1)	0 (0)	<0.001	−5.124
Difenoconazole	40.25 (23.22–97.68)	29.60 (19.24–36.17)	15 (17.6)	6 (7.1)	0.031	−2.151
Fipronil	132.85 (37.63–238.60)	ND	12 (14.1)	0 (0)	<0.001	−3.579
Imidacloprid	59.26 (43.47–108.12)	73.66 (58.43-N/A)	8 (9.4)	3 (3.5)	0.128	−1.524
Isoprothiolane	188.49 (101.76–263.86)	ND	10 (11.8)	0 (0)	0.001	−3.248
Pretilachlor	143.06 (127.29–236.12)	ND	8 (9.4)	0 (0)	0.004	−2.888
Propiconazole	186.10 (117.06–240.22)	ND	10 (11.8)	0 (0)	<0.001	−3.248
Pymetrozine	43.10 (20.96–98.91)	23.41 (8.51-N/A)	23 (27.1)	2 (2.4)	<0.001	−4.581
Tebuconazole	37.78 (20.71–72.57)	14.30 (8.26–26.78)	31 (36.5)	7 (8.2)	<0.001	−4.634
Tricyclazole	10.69 (6.34–32.50)	8.19 (6.59–17.37)	39 (45.9)	13 (15.3)	<0.001	−4.372
Trifloxystrobin	83.16 (43.40–226.32)	ND	19 (22.4)	0 (0)	<0.001	−4.601

ND: Not detected; N/A: Not available (75th percentile was not available due to small number of samples detected with target compounds); ^a^ Median was calculated based on the frequency of detection (*n*), instead of total number of samples *n* = 85.

**Table 5 ijerph-19-06806-t005:** Pesticide’s exposure and permissible exposure limit (PEL) of farmers (*n* = 85).

Target Compounds	Farmers’ Exposure (TWA) (ng/m^3^)	Permissible Exposure Limit (8-h TWA) (ng/m^3^)
Azoxystrobin	2.55	2.00 × 10^6 a^
Buprofezin	10.96	2.00 × 10^6 b^
Chlorantraniliprole	16.0	5.00 × 10^6 c^
Difenoconazole	4.88	8.00 × 10^6 a^
Fipronil	10.04	3.50 × 10^4 d^
Imidacloprid	3.35	7.00 × 10^5 e^
Isoprothiolane	10.20	5.00 × 10^6 f^
Pretilachlor	7.71	N/A
Propiconazole	10.35	8.00 × 10^6 g^
Pymetrozine	9.37	8.00 × 10^5 h^
Tebuconazole	9.82	N/A
Tricyclazole	5.75	3.00 × 10^6 i^
Trifloxystrobin	15.96	2.70 × 10^6 j^

^a^ [26]; ^b^ [27]; ^c^ [28]; ^d^ [29]; ^e^ [30]; ^f^ [31]; ^g^ [32]; ^h^ [33]; ^i^ [34]; ^j^ [35]; N/A: Not available.

**Table 6 ijerph-19-06806-t006:** Concentration of target compounds detected in blood serum samples of farmers and non-farmers.

Target Compounds	Median ^a^ (IQR)(ng/mL)		Frequency of Detection, *n* (%)		*p*-Value	*Z*-Score
	Farmers	Non-Farmers	Farmers	Non-Farmers		
Azoxystrobin	210.12 (51.22–235.58)	ND	6 (7.1)	0 (0)	0.013	−2.486
Buprofezin	75.10 (57.26–165.50)	48.73 (44.53-N/A)	10 (11.8)	2 (2.4)	0.015	−2.428
Chlorantraniliprole	73.78 (56.62–137.29)	ND	19 (22.4)	0 (0)	<0.001	−4.601
Difenoconazole	81.32 (51.46–140.39)	62.74 (56.37-N/A)	10 (11.8)	2 (2.4)	0.016	−2.400
Fipronil	75.55 (52.83–101.94)	47.83 (N/A)	18 (21.2)	1 (1.2)	<0.001	−4.156
Imidacloprid	79.96 (48.66–181.84)	48.58 (N/A)	9 (10.6)	1 (1.2)	0.009	−2.618
Isoprothiolane	91.00 (66.90–165.09)	ND	12 (14.1)	0 (0)	<0.001	−3.579
Pretilachlor	88.33 (49.08–164.82)	ND	6 (7.1)	0 (0)	0.013	−2.486
Propiconazole	105.34 (67.99–133.98)	ND	6 (7.1)	0 (0)	0.013	−2.486
Pymetrozine	124.45 (58.46–188.50)	ND	7 (8.2)	0 (0)	0.007	−2.693
Tebuconazole	79.33 (58.10–128.05)	49.66 (48.29- N/A)	15 (17.6)	2 (2.4)	<0.001	−3.373
Tricyclazole	58.27 (53.24–79.15)	55.89 (48.38-N/A)	4 (4.7)	2 (2.4)	0.402	−0.838
Trifloxystrobin	209.89 (81.28–238.62)	ND	5 (5.9)	0 (0)	0.024	−2.263

ND: Not detected; N/A: Not available (75th percentile was not available due to small number of samples detected with target compounds); ^a^ Median was calculated based on the frequency of detection (*n*), instead of total number of samples *n* = 85.

**Table 7 ijerph-19-06806-t007:** Correlation between pesticides concentration in personal air with blood serum and climatological conditions, and pesticides concentration in blood serum with BMI.

Target Compounds	Personal Air and Blood Serum				Personal Air and Climatological Conditions						Blood Serum and BMI			
	Farmers (*n* = 85)		Non-Farmers (*n* = 85)		Wind Speed		Temperature		Humidity		Farmers (*n* = 85)		Non-Farmers (*n* = 85)	
	r	*p*-Value	r	*p*-Value	r	*p*-Value	r	*p*-Value	r	*p*-Value	r	*p*-Value	r	*p*-Value
Azoxystrobin	0.727 **	<0.001	N/A	N/A	−0.043	0.696	−0.207	0.058	0.180	0.100	−0.600	0.208	N/A	N/A
Buprofezin	0.656 **	<0.001	0.494 **	<0.001	0.195	0.073	0.070	0.527	−0.125	0.256	−0.430	0.214	−1.000	N/A
Chlorantraniliprole	0.889 **	<0.001	N/A	N/A	−0.342 **	0.001	−0.463 **	<0.001	0.435 **	<0.001	0.147	0.547	N/A	N/A
Difenoconazole	0.745 **	<0.001	0.584 **	<0.001	−0.031	0.779	−0.111	0.314	0.150	0.171	0.236	0.511	−1.000	N/A
Fipronil	0.748 **	<0.001	N/A	N/A	0.048	0.665	−0.061	0.581	−0.002	0.988	−0.092	0.717	N/A	N/A
Imidacloprid	0.937 **	<0.001	0.584 **	<0.001	0.077	0.485	−0.006	0.954	−0.034	0.759	0.383	0.308	N/A	N/A
Isoprothiolane	0.917 **	<0.001	N/A	N/A	0.059	0.590	0.025	0.818	−0.093	0.397	0.224	0.484	N/A	N/A
Pretilachlor	0.868 **	<0.001	N/A	N/A	−0.023	0.834	0.039	0.726	−0.093	0.399	−0.143	0.787	N/A	N/A
Propiconazole	0.746 **	<0.001	N/A	N/A	0.252 *	0.020	0.175	0.108	−0.113	0.304	0.200	0.704	N/A	N/A
Pymetrozine	0.548 **	<0.001	N/A	N/A	0.148	0.176	0.124	0.258	−0.202	0.064	0.393	0.383	N/A	N/A
Tebuconazole	0.643 **	<0.001	0.517 **	<0.001	0.142	0.195	0.096	0.384	−0.099	0.369	−0.368	0.177	−1.000	N/A
Tricyclazole	0.367 **	<0.001	0.172	0.116	0.219 *	0.044	0.047	0.668	−0.006	0.953	0.400	0.600	−1.000	N/A
Trifloxystrobin	0.556 **	<0.001	N/A	N/A	0.129	0.241	0.095	0.390	−0.106	0.332	0.000	1.000	N/A	N/A

N/A: Not available; * *p* < 0.05; ** *p* < 0.001.

## Data Availability

All data generated or analyzed during this study are included in this published article (and its Appendix A).

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
