# Peer review of "Exposure to Airborne Pesticides and Its Residue in Blood Serum of Paddy Farmers in Malaysia"

_ijerph, 2022, doi:10.3390/ijerph19116806_

Round 1
Reviewer 1 Report
This an interesting and important paper. The English in the introduction is not as good as it should be. I would highly recommend that an editor be hired to rewrite the introduction. the inhalation aspect of the paper is first class but I worry about two other sources of contamination-oral and dermal. There will be oral contamination and it should be considered. You acknowledge that dermal contamination is a problem but you don't do any analysis of the effects of such contamination. When clothing is saturated the skin is in contact. It would be hugely interesting to understand how much of the detected compounds were absorbed thru the lungs ands how much through the skin. This is a hard problem but I should be done and you seem to have the capability of doing a proper study. I strongly encourage you to continue working on this problem. This paper isa very good start.
Reviewer 2 Report
The authors examined the concentrations of 13 CUPs in the air and in the blood of paddy farmers and non-farmers in Malaysian paddy farmlands and used UHPLC-MS/MS to analyze the results and found that the concentrations of CUPs in the air were positively correlated with the concentrations of CUPs in the blood. The contribution of this paper deserves recognition, but some problems in the article still need to be revised.
- There are formatting errors in the article such as punctuation, e.g. in line 60, " .' " should be changed to " '. "
- The formatting of Table 1 needs to be adjusted. One word shouldn’t be split into two rows,and the units need to be changed too.
- How many CUPs are currently available? Why were these 13 CUPs selected and what was the basis for their selection?
- The citation format in the main text is not ijerph, please change.
- The article recorded the BMI of the test subjects, does the fat content of the body affect the absorption of pesticides in the body and is there a relationship between BMI and pesticides in the blood?
- I wonder to know the reason for choosing men as the target group. Is it because paddy is predominantly grown by men in the area? and what percentage of the population is involved?
Round 2
Reviewer 2 Report
The comments have been amended and I think the paper can be accepted.